# Factors Associated with Cardiac/Pericardial Injury among Blunt Injury Patients: A Nationwide Study in Japan

**DOI:** 10.3390/jcm11154534

**Published:** 2022-08-03

**Authors:** Kenichiro Ishida, Yusuke Katayama, Tetsuhisa Kitamura, Tomoya Hirose, Masahiro Ojima, Shunichiro Nakao, Jotaro Tachino, Yutaka Umemura, Takeyuki Kiguchi, Tasuku Matsuyama, Tomohiro Noda, Kosuke Kiyohara, Jun Oda, Mitsuo Ohnishi

**Affiliations:** 1Department of Acute Medicine and Critical Care Medical Center, Osaka National Hospital, National Hospital Organization, Osaka 540-0006, Japan; ojimarionet999@yahoo.co.jp (M.O.); gullwing300sler@mac.com (M.O.); 2Department of Traumatology and Acute Critical Medicine, Graduate School of Medicine, Osaka University, Suita 565-0871, Japan; orion13@hp-emerg.med.osaka-u.ac.jp (Y.K.); htomoya1979@hp-emerg.med.osaka-u.ac.jp (T.H.); shunichironakao@hp-emerg.med.osaka-u.ac.jp (S.N.); jotarotachino@gmail.com (J.T.); odajun@gmail.com (J.O.); 3Division of Environmental Medicine and Population Sciences, Department of Social and Environmental Medicine, Graduate School of Medicine, Osaka University, Suita 565-0871, Japan; lucky_unatan@yahoo.co.jp; 4Department of Emergency and Critical Care, Osaka General Medical Center, Osaka 540-0006, Japan; plum0022@yahoo.co.jp (Y.U.); take_yuki888@yahoo.co.jp (T.K.); 5Kyoto University Health Service, Kyoto 606-8303, Japan; 6Department of Emergency Medicine, Kyoto Prefectural University of Medicine, Kyoto 602-8566, Japan; kame0413oka.jin@gmail.com; 7Department of Traumatology and Critical Care Medicine, Graduate School of Medicine, Osaka Metropolitan University, Osaka 558-8585, Japan; tomo-noxx@hotmail.co.jp; 8Department of Food Science, Faculty of Home Economics, Otsuma Women’s University, Tokyo 102-8357, Japan; kiyohara@otsuma.ac.jp

**Keywords:** blunt cardiac injury, cardiac contusion, sternal fractures, Japan

## Abstract

The lack of established diagnostic criteria makes diagnosing blunt cardiac injury difficult. We investigated the factors associated with blunt cardiac injury using the Japan Trauma Data Bank (JTDB) in a multicenter observational study of blunt trauma patients conducted between 2004 and 2018. The primary outcome was the incidence of blunt cardiac/pericardial injury. Multivariable logistic regression analysis was used to identify factors independently associated with blunt cardiac injuries. Of the 228,513 patients, 1002 (0.4%) had blunt cardiac injury. Hypotension on hospital arrival (adjusted odds ratio (AOR) 4.536, 95% confidence interval (CI) 3.802–5.412), thoracic aortic injury (AOR 2.722, 95% CI 1.947–3.806), pulmonary contusion (AOR 2.532, 95% CI 2.204–2.909), rib fracture (AOR 1.362, 95% CI 1.147–1.618), sternal fracture (AOR 3.319, 95% CI 2.696–4.085). and hemothorax/pneumothorax (AOR 1.689, 95% CI 1.423–2.006)) was positively associated with blunt cardiac injury. Regarding the types of patients, car drivers had a higher rate of blunt cardiac injury compared to other types of patients. Driving a car, hypotension on hospital arrival, thoracic aortic injury, pulmonary contusion, rib fracture, sternal fracture, and hemothorax/pneumothorax were positively associated with blunt cardiac injury.

## 1. Introduction

Blunt cardiac injury is a rare trauma. Several studies using large databases reported cardiac injury in 0.3–2.3% of blunt trauma cases [1,2,3]. Blunt cardiac injury can be fatal, and several studies showed that it occurs in 10–20% of trauma-related deaths [4,5,6]. For patients alive on arrival at the hospital, prompt diagnosis and treatment are critical to saving their lives [7,8].

Initial trauma management is performed according to standardized protocols such as Advanced Trauma Life Support [9]. In Japan, there is a standardized protocol called Japan Advanced Trauma Evaluation and Care (JATEC), developed by the Japan Association of Trauma Medicine [10,11]. JATEC and ATLS are essentially the same [9,10]. Symptoms such as chest pain, dyspnea, palpitations, and arrhythmias are associated with blunt cardiac injury [12,13,14]. However, these symptoms are difficult to assess in cases of severe trauma, such as hemorrhagic shock or head injury with impaired consciousness.

There have been several observational studies of the injuries associated with blunt cardiac injury, including sternal fractures, rib fractures, and hemopneumothorax [1,2,6,15,16]. The Eastern Association for the Surgery of Trauma guidelines state that “blunt cardiac injury should be considered with severe mechanism trauma unresponsive to resuscitation efforts” [17]. They also recommend a 12-lead electrocardiogram and measurement of serum troponin levels to screen for blunt cardiac injury in such cases [17]. However, the lack of established diagnostic criteria makes it difficult to diagnose blunt cardiac injury based on a single test or set of diagnostic criteria. Therefore, the blunt cardiac injury should be diagnosed based on several parameters [17,18].

Thus, identifying and verifying factors associated with blunt cardiac injury may help clinicians diagnose the blunt cardiac injury. Our study investigated factors associated with blunt cardiac injury by analyzing the Japan Trauma Data Bank (JTDB).

## 2. Materials and Methods

### 2.1. Study Design, Population, and Setting

This multicenter observational study analyzed data from patients enrolled in the JTDB between 2004 and 2018. The study was approved by the Ethics Committee of Osaka National Hospital. We did not obtain informed consent, as the data were anonymous (Approval No. 19–9). Our findings are described in accordance with the “Strengthening the Reporting of Observational Studies in Epidemiology (STROBE) Statement for Observational Studies” [19].

### 2.2. Japan Trauma Data Bank

The JTDB is a hospital-based trauma registry in Japan maintained by the Japanese Association of Emergency Medicine and the Japanese Society of Trauma Medicine [20]. During the study period, patients were enrolled from 280 facilities equivalent to level 1 trauma centers in the United States [11]. Since JTDB has been registered using the Abbreviated Injury Scale (AIS) version 1998 during the study periods. Thus, online AIS version 1998 data were collected from participating facilities. Physicians and medical assistants who had completed the AIS coding course registered the data in most cases. For registration in the JTDB, an AIS score > 2 is required. In total, 92 elements of information are registered in the JTDB, including age, sex, mechanism of injury, type of transportation, vital signs, AIS code for each injured body site, the Injury Severity Score (ISS), procedures performed in the hospital, and hospital outcome. The findings of this study do not represent the official opinion of the Japanese Association of Emergency Medicine or the Japanese Association of Trauma Medicine.

### 2.3. Participants

Blunt trauma patients were included in the study. We excluded cases with interhospital transport, cardiopulmonary arrest on arrival (CPA), or missing data for sex, vital signs, ISS, or hospital discharge.

### 2.4. Variables 

The following JTDB data were analyzed in this study: age, sex, type of patient systolic blood pressure on arrival, heart rate on arrival, thoracic injury for which AIS > 2, concomitant non-thoracic injuries (head, facial, neck, abdominal, spinal, lower extremity (including pelvis) or upper extremity injury for which AIS > 2), concomitant thoracic injury (thoracic aortic injury, esophageal injury, pulmonary contusion, rib fracture, tracheal injury, rib fracture, sternal fracture, clavicle fracture, hemothorax/pneumothorax, thoracic spinal cord injury, or fracture), and ISS. In addition, we extracted the AIS codes for blunt cardiac/pericardial injury (Appendix A) and concomitant thoracic injury (Appendix A). In JTDB, we need to determine the presence/absence of CPAOA using vital signs at hospital arrival. Therefore, we defined CPA as systolic blood pressure and/or pulse of 0 on arrival at the hospital [21]. Hypotension was defined as systolic blood pressure < 80 mmHg on arrival at the hospital [22,23].

### 2.5. Outcome Measures

The primary outcome was the incidence of blunt cardiac/pericardial injury. Secondary outcomes included the incidence of procedures (emergency pericardiocentesis, emergency pericardiotomy, emergency thoracotomy, and cardiac surgery), death in the emergency room (ER), in-hospital mortality, and complications during hospitalization (acute kidney injury, acute respiratory distress syndrome, myocardial infarction, pulmonary embolism, and pneumonia).

### 2.6. Statistical Analysis

We divided the eligible patients into two groups according to the presence/absence of cardiac/pericardial injury. First, we generated descriptive statistics for these two groups, i.e., median and interquartile range (IQR) for continuous variables and frequency and percentage for categorical variables. We used the Mann–Whitney U-test for comparing the groups in terms of continuous variables and the χ2 or Fisher’s exact test for comparing them in terms of categorical variables.

Using the forced entry method, multivariate logistic regression analysis was performed to obtain adjusted odds ratios (AORs) and 95% confidence intervals (CI) for factors associated with blunt cardiac injury. We included the following factors: admission year; age; sex; type of patient; hypotension on arrival; head, facial, neck, abdominal, spinal, lower extremity (including pelvis), upper extremity for which AIS > 2 (presence or absence); and thoracic aortic injury, pulmonary contusion, rib fracture, sternal fracture, clavicle fracture, hemothorax/pneumothorax, or thoracic spinal cord injury/fracture (presence or absence). These variables were selected based on previous studies and according to whether they are considered clinically relevant to cardiac/pericardial injury [1,2,3,16,17]. 

Traffic accidents account for a high percentage of blunt trauma and blunt cardiac injuries [1,2,24]. Thus, performed a subgroup multivariate logistic regression analysis of traffic accident patients (car drivers, front seat passengers, back seat passengers, motorcyclists, bicyclists, and pedestrians). We grouped the patients according to the AIS score of the blunt cardiac injury, with consideration of procedure, prognosis, and complications. We used the Cochran–Armitage test for analyzing categorical variables. The two-sided significance level for all tests was *p* < 0.05. We used R software (version 4.0.3; https://www.r-project.org/ (accessed on 26 July 2022)) for all statistical analyses.

## 3. Results

Figure 1 shows the patient flow in this study. During the study period, 354,608 patients were registered in the JTDB, including 326,058 with blunt trauma. Of the 228,513 patients, 0.4% had a blunt cardiac injury. Table 1 shows the characteristics of the eligible patients. In the cardiac injury group, the median age was 49 years, and 70.1% were male. The most common type of patient was car driver, followed by motorcyclist. The median systolic blood pressure on arrival was 114 mmHg, and the proportion of patients with hypotension on arrival was 20.7%. The median ISS was 25. The most common concomitant injury was a head injury. The most common concomitant thoracic injury was rib fracture, followed by pulmonary contusion. The most common AIS score for blunt cardiac injury was 1, followed by 5.

Table 2 shows the associations between blunt cardiac injury and various factors. In multivariate logistic analysis, age < 20 years, hypotension on arrival, abdominal injury, thoracic aortic injury, pulmonary contusion, rib fracture, sternal fracture, and hemothorax/pneumothorax were positively associated with blunt cardiac injury. On the other hand, head injury, spinal injury, and clavicle fractures were inversely associated with blunt cardiac injury. Regarding admission year, 2013–2015 and 2016–2018 were inversely associated with blunt cardiac injury, unlike 2004–2006. Regarding the type of patient, front seat passenger, back seat passenger, motorcyclist, bicyclist, pedestrian, fall from a height, and falling down were inversely associated with blunt cardiac injury, unlike car driver.

Table 3 shows the results of the subgroup analysis of patients with traffic accident injuries. Multivariate logistic regression analysis showed that hypotension on arrival, thoracic aortic injury, pulmonary contusion, rib fracture, sternal fracture, and hemothorax/pneumothorax were positively associated with blunt cardiac injury. Regarding admission year, 2013–2015 and 2016–2018 were inversely associated with blunt cardiac injury, unlike 2004–2006. Regarding the type of patient, front seat passenger, back seat passenger, motorcyclist, bicyclist, and pedestrian were inversely associated with blunt cardiac injury, unlike car driver.

Table 4 shows the results for the other outcomes. Emergency pericardiocentesis, emergency pericardiotomy, emergency thoracotomy, and cardiac surgery were performed in 9.9%, 6.4%, 23.5%, and 11.6% of cases of blunt cardiac injury, respectively. The in-hospital mortality rate for blunt cardiac injury was 23.3%. Higher AIS scores were associated with higher rates of procedures and in-hospital mortality. The most common complication was pneumonia, followed by acute respiratory distress syndrome.

## 4. Discussion

Our study investigated factors associated with blunt cardiac injury in blunt trauma cases using the JTDB. Blunt cardiac injury occurred in 0.4% of blunt trauma patients. Multivariate logistic regression analysis showed that car driver, hypotension on hospital arrival, thoracic aortic injury, pulmonary contusion, rib fracture, sternal fracture, and hemothorax/pneumothorax were positively associated with blunt cardiac injury. We identified factors associated with blunt cardiac injury through analysis of a large database, which may help clinicians diagnose cardiac injury based on the injury mechanism and coexisting injuries.

First, our results showed that being a car driver was more strongly associated with blunt cardiac injury compared to the other types of patients. Traffic accidents account for high percentages of blunt trauma and blunt cardiac injuries [1,2,5,24]. In Japan, the number of traffic accidents and traffic fatalities has decreased [25,26]. In our study, traffic accidents accounted for a high proportion of blunt cardiac injuries. The incidence of blunt cardiac injury has decreased over time in patients involved in traffic accidents. This result was similar to a previous study in the United States [3]. The decrease in the incidence of cardiac injury could be attributed to changes in the environment surrounding patients with injuries, such as the decrease in hazardous driving following the revision of the Road Traffic Law and the widespread development and application of advanced technology [27], such as airbag technology [28]. Mechanisms for blunt cardiac injury include compression of the heart by the sternum or spine and a mechanism involving deceleration [8,29]. In our study, car drivers were more likely to suffer a blunt cardiac injury in traffic accidents compared to other types of patients; car drivers may be struck in the chest by an airbag or the steering wheel due to the sudden deceleration associated with traffic accidents. Prior research suggests that handling deformity is an important indicator of thoracic injury [30]. Although air bags disperse the impact over the chest, thoracic trauma may occur if the impact is sufficient to crush the air bag. Even if the air bag is adequate for absorbing the initial impact, it is unlikely to be effective for subsequent events, such as vehicle rollover. The widespread use of airbags in motor vehicles is thought to have contributed to the reductions in traffic fatalities and blunt cardiac injuries. However, blunt thoracic and cardiac injuries can still occur even when a car is equipped with airbags [28,30,31].

Second, our results showed that about 20% of patients had hypotension on arrival at the hospital, and hypotension was positively associated with blunt cardiac injury. Hypotension in the context of trauma is classified as a hypovolemic shock due to hemorrhage, obstructive shock due to tension pneumothorax or cardiac tamponade, cardiogenic shock due to cardiac injury, or distributive shock due to spinal cord injury [32]. Regarding the pathophysiology of these shocks, in patients with blunt cardiac injury, hemorrhage, obstruction, and cardiogenic pathophysiology lead to hypotension. Thus, blunt cardiac injury and hypotension on arrival may be related. Suspicion of blunt cardiac injury based on several factors, such as the mechanism of injury, hemodynamics, and comorbid injuries, may contribute to the diagnosis of blunt cardiac injury.

Third, in our multivariate logistic analysis of the JTDB, thoracic aortic injury, pulmonary contusion, rib fracture, sternal fracture, and hemothorax/pneumothorax were positively associated with blunt cardiac injury. Sternal fractures were strongly associated with blunt cardiac injury. Previous studies reported sternal fractures in approximately 2% of blunt trauma cases [33,34], while in 2.4–8% of patients with sternal fractures, blunt cardiac injury were also observed [33,35,36,37]. Observational studies in the United States and Germany using large databases have also reported that sternal fracture is associated with blunt cardiac injury [1,2]. In our Japanese database analysis, sternal fracture was found in 12.7% of blunt cardiac injuries, and logistic regression analysis showed that these factors were associated with each other. Although computed tomography (CT) may not be feasible in cases of hemodynamic instability, it is necessary for an accurate diagnosis of sternal fracture [33]. The diagnostic utility of ultrasound for sternal fracture has also been reported [38,39]. The sternal fracture itself is a minor injury; however, when a strong external force causes a sternal fracture, the heart may be injured by that force. In addition to sternal fractures, thoracic aortic injuries, pulmonary contusions, rib fractures, and hemothorax/pneumothorax were positively associated with blunt cardiac injury in this study, similar to previous studies [1,2]. Diagnosis of these injuries requires measurement of troponin levels and a 12-lead electrocardiogram.

Fourth, the mortality rate for blunt cardiac injury ranged from 13 to 89% [1,3,7,8,16,24,40]. A possible reason for differences in mortality rates among studies is that the diagnostic criteria for cardiac injury, and injury severity, vary. As shown in Table 4, the overall rate of death due to blunt cardiac injury in the ER was 9.6%, and the in-hospital mortality rate was 23.3%. In addition, blunt cardiac injuries with high AIS scores tended to be treated via emergency pericardiocentesis, emergency thoracotomy, or cardiac surgery. The mortality rate rose sharply with the severity of blunt cardiac injury (i.e., the AIS score). These results suggest that prompt severity assessment and treatment are needed, rather than a mere diagnosis of blunt cardiac injury.

There were several limitations to our study. First, although we used Japanese nationwide data in our analysis, JTDB is not population-based data. The changes in the number of cardiac injuries were influenced by the number of facilities enrolled in JTDB. Second, blunt cardiac injury patients were diagnosed and registered at the institutions that they presented to, and the lack of established diagnostic criteria for cardiac injury could have resulted in inaccurate reporting. Serum troponin levels, the presence of arrhythmias, and echocardiographic results are considered relevant for the diagnosis of blunt cardiac injury, but we did not obtain these data from the JTDB. In addition, if a severe heart injury is diagnosed and treated only based on a chest X-ray or ultrasound, other minor injuries may be missed. Third, data on the car model, airbag status, and seatbelt use were unavailable from the JTDB for car drivers. Fourth, the causes of hypotension are not available in the JTDB. Fifth, the AIS codes for sternal fractures were extracted from the JTDB, but fracture details (close vs. open, displaced vs. non-displaced) were unavailable. Previous studies have shown that most sternal fractures associated with blunt cardiac injury are closed fractures. However, open fractures show a stronger association with blunt cardiac injury than closed fractures [2]. The associations between such fracture parameters and blunt cardiac injury cannot be investigated when using the JTDB. Finally, this was an observational study, and there may have been unidentified confounding factors. Despite these limitations, the strengths of our study are its large sample size and time trends, which provides important generalizability for the incidence and associated factors of blunt cardiac injury.

## 5. Conclusions

In this study, being a car driver, hypotension on hospital arrival, thoracic aortic injury, pulmonary contusion, rib fracture, sternal fracture, and hemothorax/pneumothorax were positively associated with blunt cardiac injury. Therefore, screening for cardiac injury is required when these injuries are diagnosed, and careful observation should be continued after hospitalization. If clinicians suspect cardiac injury from such concomitant injuries, diagnosing and grading cardiac injury more promptly may be possible. This may contribute to improved patient prognosis.

## Figures and Tables

**Figure 1 jcm-11-04534-f001:**
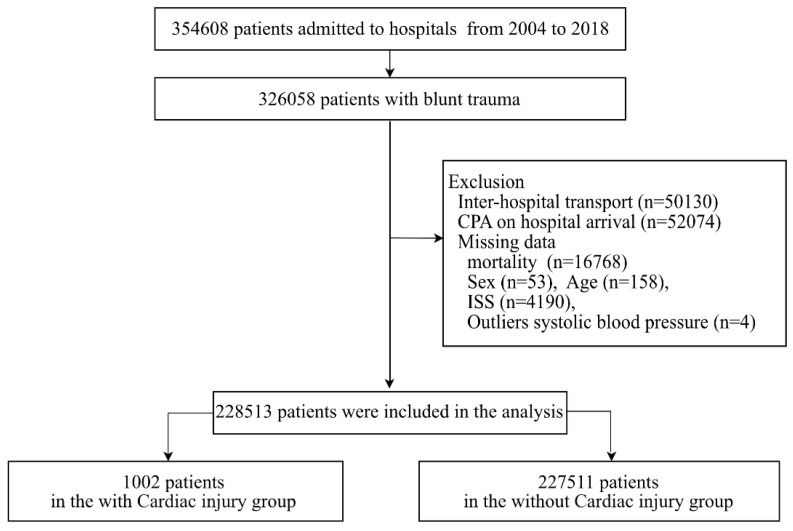
Flowchart of patient selection from the Japan Trauma Data Bank.

**Table 1 jcm-11-04534-t001:** Characteristics of the cardiac injury and non-cardiac injury groups.

	Cardiac Injury (+)	Cardiac Injury (−)	
	N = 1002	N = 227,511	*p* Value
Admission year, *n* (%)			<0.001
2004–2006	51 (5.1)	7612 (3.4)	
2007–2009	171 (17.1)	24,145 (10.6)	
2010–2012	276 (27.5)	49,482 (21.8)	
2013–2015	270 (27.0)	77,749 (34.2)	
2016–2018	234 (23.4)	68,523 (30.1)	
Age, years, median (IQR)	49 [27–68]	62 [38–78]	<0.001
Male, sex, *n* (%)	702 (70.1)	141,711 (62.3)	<0.001
Type of patients, *n* (%)			<0.001
Car driver	313 (31.2)	20,272 (8.9)	
Front seat passenger	48 (4.8)	3879 (1.7)	
Back seat passenger	24 (2.4)	2952 (1.3)	
Motorcyclist	193 (19.3)	27,425 (12.1)	
Bicyclist	59 (5.9)	19,098 (8.4)	
Pedestrian	94 (9.4)	18,368 (8.1)	
Fall from heights	128 (12.8)	21,631 (9.5)	
Fall down	39 (3.9)	91,036 (40.0)	
Other	104 (10.4)	22,850 (10.0)	
Systolic BP on arrival, mmHg, median (IQR)	114 [83–139]	137 [137–158]	<0.001
Heart rate on arrival, bpm, median (IQR)	96 [80–116]	83 [72–96]	<0.001
Hypotension (SBP < 80 mmHg), *n* (%)	207 (20.7)	7441 (3.3)	<0.001
Concomitant injury			
Head injury, *n* (%)	238 (23.8)	72,207 (31.7)	<0.001
Face injury, *n* (%)	11 (1.1)	1805 (0.8)	0.366
Neck injury, *n* (%)	1 (0.1)	374 (0.2)	0.910
Chest injury, *n* (%)	824 (82.2)	49,037 (21.6)	<0.001
Abdominal injury, *n* (%)	160 (16.0)	10,550 (4.6)	<0.001
Spine injury, *n* (%)	75 (7.5)	23,976 (10.5)	0.002
Lower extremity injury including pelvis, *n* (%)	237 (23.7)	64,644 (28.4)	0.001
Upper extremity injury, *n* (%)	52 (5.2)	12,202 (5.4)	0.863
Concomitant thoracic injury			
Thoracic aortic injury, *n* (%)	41 (4.1)	1069 (0.5)	<0.001
Esophageal injury, *n* (%)	0 (0.0)	23 (0.0)	1.000
Pulmonary contusion, *n* (%)	420 (41.9)	25,084 (11.0)	<0.001
Tracheal injury, *n* (%)	2 (0.2)	108 (0.0)	0.142
Rib fracture, *n* (%)	483 (48.2)	42,585 (18.7)	<0.001
Sternal fracture, *n* (%)	127 (12.7)	3675 (1.6)	<0.001
Clavicle fracture, *n* (%)	91 (9.1)	12,240 (5.4)	<0.001
Hemothorax/pneumothorax, *n* (%)	372 (37.1)	26,893 (11.8)	<0.001
Thoracic spinal cord injury/fracture, *n* (%)	66 (6.6)	11,105 (4.9)	0.015
AIS of cardiac injury, *n* (%)			
1	594 (59.3)	−	
2	43 (4.3)	−	
3	88 (8.8)	−	
4	20 (2.0)	−	
5	224 (22.4)	−	
6	33 (3.3)	−	
ISS, median (IQR)	25 [17–35]	10 [9–19]	<0.001

IQR—interquartile range; BP—blood pressure; AIS—Abbreviated Injury Scale.

**Table 2 jcm-11-04534-t002:** Univariable and multivariable logistic regression to identify factors associated with cardiac injury among blunt trauma patients.

	Cardiac Injury	Univariable Analyses		Multivariable Analyses	
N = 228,513	% (*n*/N)	Crude OR (95% CI)	*p* Value	Adjusted ^a^ OR (95% CI)	*p* Value
Admission year					
2004–2006	0.67 (51/7663)	Reference		Reference	
2007–2009	0.70 (171/24,316)	1.057 (0.772–1.447)	0.729	1.167 (0.849–1.604)	0.340
2010–2012	0.55 (276/49,758)	0.833 (0.617–1.123)	0.231	1.058 (0.780–1.434)	0.716
2013–2015	0.35 (270/78,019)	0.518 (0.384–0.700)	<0.001	0.719 (0.530–0.976)	0.034
2016–2018	0.34 (234/68,757)	0.510 (0.376–0.691)	<0.001	0.711 (0.521–0.970)	0.032
Age groups					
<20 years	0.54 (123/22,798)	0.936 (0.770–1.138)	0.507	1.323 (1.076–1.626)	0.008
20–64 years	0.58 (575/99,788)	Reference		Reference	
≧65 years	0.29 (304/105,927)	0.497 (0.432–0.571)	<0.001	1.029 (0.885–1.196)	0.713
Sex					
Male	0.49 (702/142,413)	1.417 (1.237–1.622)	<0.001	0.968 (0.838–1.118)	0.657
Female	0.35 (300/86,100)	Reference		Reference	
Type of patients					
Car driver	1.52 (313/20,585)	Reference		Reference	
Front seat passenger	1.22 (48/3927)	0.801 (0.590–1.088)	0.156	0.722 (0.526–0.991)	0.044
Back seat passenger	0.81 (24/2976)	0.527 (0.347–0.799)	0.003	0.554 (0.361–0.850)	0.007
Motorcyclist	0.70 (193/27,618)	0.456 (0.381–0.546)	<0.001	0.534 (0.441–0.647)	<0.001
Bicyclist	0.31 (59/19,157)	0.200 (0.151–0.264)	<0.001	0.311 (0.232–0.415)	<0.001
Pedestrian	0.51 (94/18,462)	0.331 (0.263–0.418)	<0.001	0.393 (0.307–0.502)	<0.001
Fall from heights	0.59 (128/21,759)	0.383 (0.312–0.471)	<0.001	0.390 (0.313–0.486)	<0.001
Fall down	0.04 (39/91,075)	0.028 (0.020–0.039)	<0.001	0.058 (0.041–0.082)	<0.001
Other	0.45 (104/22,954)	0.295 (0.236–0.368)	<0.001	0.421 (0.335–0.530)	<0.001
Hypotension on arrival					
(+)	2.71 (207/7648)	7.701 (6.597–8.989)	<0.001	4.536 (3.802–5.412)	<0.001
(−)	0.36 (795/220,865)	Reference		Reference	
Head injury					
(+)	0.33 (238/72,445)	0.670 (0.579–0.775)	<0.001	0.692 (0.593–0.807)	<0.001
(−)	0.49 (764/156,068)	Reference		Reference	
Face injury					
(+)	0.61 (11/1816)	1.388 (0.765–2.519)	0.281	1.100 (0.601–2.014)	0.758
(−)	0.44 (991/226,697)	Reference		Reference	
Neck injury					
(+)	0.27 (1/375)	0.607 (0.085–4.322)	0.618	0.362 (0.050–2.612)	0.314
(−)	0.44 (1001/228,138)	Reference		Reference	
Abdominal injury					
(+)	1.49 (160/10,710)	3.908 (3.296–4.633)	<0.001	1.192 (0.991–1.433)	0.063
(−)	0.39 (842/217,803)	Reference		Reference	
Spine injury					
(+)	0.31 (75/24,051)	0.687 (0.543–0.869)	0.002	0.671 (0.523–0.861)	0.002
(−)	0.45 (927/204,462)	Reference		Reference	
Lower extremity injury including pelvis					
(+)	0.37 (237/64,881)	0.781 (0.675–0.903)	<0.001	0.896 (0.767–1.047)	0.167
(−)	0.47 (765/163,632)	Reference		Reference	
Upper extremity injury					
(+)	0.42 (52/12,254)	0.966 (0.730–1.278)	0.808	0.844 (0.635–1.122)	0.243
(−)	0.44 (950/216,259)	Reference		Reference	
Thoracic aortic injury					
(+)	3.69 (41/1110)	9.037 (6.574–12.424)	<0.001	2.722 (1.947–3.806)	<0.001
(−)	0.42 (961/227,403)	Reference		Reference	
Pulmonary contusion					
(+)	1.65 (420/25,504)	5.824 (5.133–6.607)	<0.001	2.532 (2.204–2.909)	<0.001
(−)	0.29 (582/203,009)	Reference		Reference	
Rib fracture					
(+)	1.12 (483/43,068)	4.041 (3.569–4.576)	< 0.001	1.362 (1.147–1.618)	<0.001
(−)	0.28 (519/185,445)	Reference		Reference	
Sternal fracture					
(+)	3.34 (127/3802)	8.840 (7.318–10.679)	< 0.001	3.319 (2.696–4.085)	<0.001
(−)	0.39 (875/224,711)	Reference		Reference	
Clavicle fracture					
(+)	0.74 (91/12,331)	1.757 (1.415–2.181)	<0.001	0.792 (0.631–0.992)	0.043
(−)	0.42 (911/216,182)	Reference		Reference	
Hemothorax/pneumothorax					
(+)	1.36 (372/27,265)	4.405 (3.873–5.010)	<0.001	1.689 (1.423–2.006)	<0.001
(−)	0.31 (630/201,248)	Reference		Reference	
Thoracic spinal cord injury/fracture					
(+)	0.59 (66/11,171)	1.374 (1.070–1.765)	0.013	0.828 (0.633–1.084)	0.170
(−)	0.43 (936/217,342)	Reference		Reference	

OR—odds ratio; IQR—interquartile range; ISS—Injury Severity Score. ^a^ Adjusted for admission year, age, sex, type of patients, hypotension on arrival, head injury, facial injury, neck injury, abdominal injury, spinal injury, lower extremity injury including pelvis, upper extremity injury, thoracic aortic injury, pulmonary contusion, rib fracture, sternal fracture, clavicle fracture, hemothorax/pneumothorax, and thoracic spinal cord injury/fracture.

**Table 3 jcm-11-04534-t003:** Univariable and multivariable logistic regression to identify factors associated with cardiac injury among traffic accident patients (subgroup analysis).

	Cardiac Injury	Univariable Analyses		Multivariable Analyses	
N = 65,107	% (*n*/N)	Crude OR (95% CI)	*p* Value	Adjusted ^a^ OR (95% CI)	*p* Value
Admission year					
2004–2006	0.98 (43/4368)	Reference		Reference	
2007–2009	1.04 (124/11,933)	1.056 (0.745–1.497)	0.759	1.016 (0.714–1.446)	0.931
2010–2012	0.97 (203/21,028)	0.980 (0.704–1.365)	0.907	0.955 (0.683–1.336)	0.787
2013–2015	0.63 (191/30,505)	0.634 (0.455–0.884)	0.007	0.618 (0.441–0.866)	0.005
2016–2018	0.68 (170/24,891)	0.692 (0.494–0.968)	0.032	0.642 (0.455–0.905)	0.011
Age groups					
<20 years	0.63 (88/14,035)	0.783 (0.621–0.986)	0.038	1.235 (0.967–1.577)	0.090
20–64 years	0.8 (411/51,402)	Reference		Reference	
≧65 years	0.85 (232/27,288)	1.064 (0.905–1.250)	0.453	1.155 (0.969–1.376)	0.108
Sex					
Male	0.78 (495/63,157)	0.982 (0.840–1.147)	0.817	0.873 (0.738–1.032)	0.111
Female	0.8 (236/29,568)	Reference		Reference	
Type of patients					
Car driver	1.52 (313/20,585)	Reference		Reference	
Front seat passenger	1.22 (48/3927)	0.801 (0.590–1.088)	0.156	0.711 (0.517–0.976)	0.035
Back seat passenger	0.81 (24/2976)	0.527 (0.347–0.799)	0.003	0.533 (0.347–0.819)	0.004
Motorcyclist	0.7 (193/27,618)	0.456 (0.381–0.546)	<0.001	0.535 (0.441–0.649)	<0.001
Bicyclist	0.31 (59/19,157)	0.200 (0.151–0.264)	<0.001	0.289 (0.216–0.388)	<0.001
Pedestrian	0.51 (94/18,462)	0.331 (0.263–0.418)	<0.001	0.360 (0.280–0.462)	<0.001
Hypotension on arrival					
(+)	3.7 (137/3704)	5.718 (4.734–6.905)	<0.001	4.369 (3.529–5.408)	<0.001
(−)	0.67 (594/89,021)	Reference		Reference	
Head injury					
(+)	0.55 (179/32,658)	0.594 (0.502–0.704)	<0.001	0.720 (0.601–0.862)	<0.001
(−)	0.92 (552/60,067)	Reference		Reference	
Face injury					
(+)	0.81 (8/991)	1.024 (0.509–2.062)	0.946	1.145 (0.563–2.326)	0.709
(−)	0.79 (723/91,734)	Reference		Reference	
Neck injury					
(+)	0.55 (1/183)	0.691 (0.097–4.934)	0.713	0.524 (0.072–3.801)	0.523
(−)	0.79 (730/92,542)	Reference		Reference	
Abdominal injury					
(+)	1.76 (122/6944)	2.501 (2.056–3.043)	<0.001	1.186 (0.959–1.466)	0.116
(−)	0.71 (609/85,781)	Reference		Reference	
Spine injury					
(+)	0.44 (33/7526)	0.533 (0.376–0.757)	<0.001	0.490 (0.341–0.705)	<0.001
(−)	0.82 (698/85,199)	Reference		Reference	
Lower extremity injury including pelvis					
(+)	0.8 (162/20,346)	1.013 (0.850–1.207)	0.886	0.920 (0.765–1.106)	0.376
(−)	0.79 (569/72,379)	Reference		Reference	
Upper extremity injury					
(+)	0.81 (35/4309)	1.032 (0.734–1.451)	0.856	0.915 (0.646–1.295)	0.615
(−)	0.79 (696/88,416)	Reference		Reference	
Thoracic aortic injury					
(+)	4.22 (30/711)	5.738 (3.951–8.334)	<0.001	2.830 (1.914–4.184)	<0.001
(−)	0.76 (701/92,014)	Reference		Reference	
Pulmonary contusion					
(+)	1.87 (303/16,246)	3.377 (2.912–3.916)	<0.001	2.427 (2.068–2.847)	<0.001
(−)	0.56 (428/76,479)	Reference		Reference	
Rib fracture					
(+)	1.44 (341/23,615)	2.582 (2.231–2.988)	<0.001	1.260 (1.035–1.532)	0.021
(−)	0.56 (390/69,110)	Reference		Reference	
Sternal fracture					
(+)	3.37 (95/2818)	4.897 (3.934–6.095)	<0.001	2.715 (2.138–3.447)	<0.001
(−)	0.71 (636/89,907)	Reference		Reference	
Clavicle fracture					
(+)	0.82 (70/8557)	1.042 (0.814–1.334)	0.745	0.756 (0.584–0.978)	0.033
(−)	0.79 (661/84,168)	Reference		Reference	
Hemothorax/pneumothorax					
(+)	1.77 (248/14,011)	2.919 (2.502–3.405)	<0.001	1.591 (1.303–1.942)	<0.001
(−)	0.61 (483/78,714)	Reference		Reference	
Thoracic spinal cord injury/fracture					
(+)	0.87 (32/3694)	1.104 (0.774–1.576)	0.585	0.809 (0.557–1.175)	0.267
(−)	0.79 (699/89,031)	Reference		Reference	

AIS—Abbreviated Injury Scale; OR—odds ratio; IQR—interquartile range; ISS—Injury Severity Score. ^a^ Adjusted for admission year, age, sex, type of patients, hypotension on arrival, head injury, facial injury, neck injury, abdominal injury, spinal injury, lower extremity injury including pelvis, upper extremity injury, thoracic aortic injury, pulmonary contusion, rib fracture, sternal fracture, clavicle fracture, hemothorax/pneumothorax, and thoracic spinal cord injury/fracture.

**Table 4 jcm-11-04534-t004:** Procedures and prognosis following arrival at the hospital among cardiac injury patients.

	Overall	AIS 1	AIS 2	AIS 3	AIS 4	AIS 5	AIS 6	*p* for Trend
	N = 1002	N = 594	N = 43	N = 88	N = 20	N = 224	N = 33	
Procedure, *n* (%)								
Emergency pericardiocentesis	99 (9.9)	7 (1.2)	4 (9.3)	19 (21.6)	1 (5.0)	60 (26.8)	8 (24.2)	<0.001
Emergency pericardial incision	64 (6.4)	1 (0.2)	2 (4.7)	13 (14.8)	6 (30.0)	34 (15.2)	8 (24.2)	<0.001
Emergency thoracotomy	235 (23.5)	22 (3.7)	7 (16.3)	26 (29.5)	11 (55.0)	147 (65.6)	22 (66.7)	<0.001
Cardiac operation	116 (11.6)	0 (0.0)	1 (2.3)	5 (5.7)	0 (0.0)	100 (44.6)	10 (30.3)	<0.001
Prognosis, *n* (%)								
Death in the ER	96 (9.6)	8 (1.3)	1 (2.3)	10 (11.4)	6 (30.0)	55 (24.6)	16 (48.5)	<0.001
In-hospital mortality	233 (23.3)	44 (7.4)	10 (23.3)	30 (34.1)	14 (70.0)	108 (48.2)	27 (81.8)	<0.001
Complication, *n* (%)								
Acute kidney injury	4 (0.4)	0 (0.0)	2 (4.7)	0 (0.0)	0 (0.0)	2 (0.9)	0 (0.0)	0.192
Acute respiratory distress syndrome	7 (0.7)	3 (0.5)	1 (2.3)	1 (1.1)	0 (0.0)	1 (0.4)	1 (3.0)	0.575
Myocardial infarction	1 (0.1)	0 (0.0)	0 (0.0)	0 (0.0)	0 (0.0)	1 (0.4)	0 (0.0)	0.136
Pulmonary embolism	2 (0.2)	0 (0.0)	1 (2.3)	0 (0.0)	0 (0.0)	0 (0.0)	1 (3.0)	0.187
Pneumonia	29 (2.9)	17 (2.9)	3 (7.0)	4 (4.5)	1 (5.0)	4 (1.8)	0 (0.0)	0.354
ER, emergency room								

## Data Availability

The data that support the findings of this study are available from the JTDB, although the availability of these data is restricted.

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
