# Peer review of "Factors Associated with Cardiac/Pericardial Injury among Blunt Injury Patients: A Nationwide Study in Japan"

_jcm, 2022, doi:10.3390/jcm11154534_

Round 1

Reviewer 1 Report

Thanks for the possibility to review this manuscript. This paper describes the incidence and factor related with cardiac and pericardial injuries in blunt trauma. This study is maked with a large trauma registry in Japan from 2014 to 2018. Main finding are describe the incidence of pericardial and cardiac injuries in blunt trauma and describe risk factors and treatment maneuvers related with this disease.

The strengths of this study are the sample size and the time trends. Registry studies are especially interesting in rare diseases, such as cardiac and pericardial injuries in blunt trauma

The weaknesses of this study are:

• The writing of the introduction is unstructured and repetitive.

• Some sentences in the introduction are inappropriate, eg 48-49 lines related with ATLS. 65-66 lines related to the use of Japan registration.

• The introduction should review this topic and conclude on the need for this work, with the appropriate bibliography. This aspect can be improved.

• Some aspects of the method are not necessary, for example, the definition of cardiopulmonary arrest.

·       The variables used for this study are described in many places in this document. Not necessary in the opinion of this reviewer. • The AIS version is old. Please justify it. • The discussion is a repetition of the results and does not relate to the appropriate bibliography. Another wording would be necessary and relate it to the bibliography on the subject. • Some limitations of this article are correctly pointed out by the authors, such as auxiliary tests for the diagnosis of blunt cardiac and pericardial injuries (biomarkers, echo, presence of arrhythmias, forensic report, etc). These are very important limitations and reduce interest in this article. • Conclusions are repetitions of the results. • The results are expressed in a confusing and cumbersome way. Tables and text are repetitive. In the opinion of this reviewer, this point of this manuscript should be improved.

In general, the manuscript has many limitations for its publication. It is not clear in its format, it is not structured and it is repetitive. More literature related to this topic is needed and the appropriate literature needs to be combined with the results.

Reviewer 2 Report

I congratulate the authors for this thorough research paper. It is well written, full of very detailed and informative data.

I did enjoy reading it very much.

I only have a few questions, that the authors could address:

- I'd like some precision about the choice of SBP under 80 for hypotension, as classicaly ATLS would define it as < 90mmHg. Maybe, among the citations reported in the method section (1/3, 13, 16) a specific one defining the chosen threshold.

- Regarding the correlation or inverse correlation with the years of admission, I wonder if the authors could infer the reason why earlier year (2004-2006) had more cardiac injuries: different reporting, traffic accident type, change in airbag technology...? Maybe a sentence could be added in the discussion.

- Regarding the discussion or the conclusion: could the authors make a stronger suggestions on how to us the generated information in the practice? Should an echocardiography or other radiological exam be systematically done in certain cases, beyond the initial trauma assessment? What about a stratifed severity of cardiac injury, in correlation with other chest injuries? Could perspectives for research studies, regarding monitoring or support strategies, be drawn from the presented data?
